# Revisiting Fur Regulon Leads to a Comprehensive Understanding of Iron and Fur Regulation

**DOI:** 10.3390/ijms24109078

**Published:** 2023-05-22

**Authors:** Chaofan Hou, Lin Liu, Xian Ju, Yunzhu Xiao, Bingyu Li, Conghui You

**Affiliations:** Shenzhen Key Laboratory of Microbial Genetic Engineering, College of Life Sciences and Oceanology, Shenzhen University, Shenzhen 518055, China; 1800251003@email.szu.edu.cn (C.H.); 2060251041@email.szu.edu.cn (L.L.); yc27690@umac.mo (X.J.); xiaoyunzhu8891@163.com (Y.X.); byli1988@foxmail.com (B.L.)

**Keywords:** ferric-uptake regulator, iron metabolism, carbon metabolism, respiration, motility, *Escherichia coli*

## Abstract

Iron is an essential element because it functions as a cofactor of many enzymes, but excess iron causes cell damage. Iron hemostasis in *Escherichia coli* was transcriptionally maintained by the ferric uptake regulator (Fur). Despite having been studied extensively, the comprehensive physiological roles and mechanisms of Fur-coordinated iron metabolism still remain obscure. In this work, by integrating a high-resolution transcriptomic study of the Fur wild-type and knockout *Escherichia coli* K-12 strains in the presence or absence of iron with high-throughput ChIP-seq assay and physiological studies, we revisited the regulatory roles of iron and Fur systematically and discovered several intriguing features of Fur regulation. The size of the Fur regulon was expanded greatly, and significant discrepancies were observed to exist between the regulations of Fur on the genes under its direct repression and activation. Fur showed stronger binding strength to the genes under its repression, and genes that were repressed by Fur were more sensitive to Fur and iron regulation as compared to the genes that were activated by Fur. Finally, we found that Fur linked iron metabolism to many essential processes, and the systemic regulations of Fur on carbon metabolism, respiration, and motility were further validated or discussed. These results highlight how Fur and Fur-controlled iron metabolism affect many cellular processes in a systematic way.

## 1. Introduction

Iron is an essential nutrient to all kingdoms of life. In the form of isolated iron or an iron complex such as a heme group or Fe-S cluster, iron functions as a cofactor by incorporating into a large number of enzymes working in pathways ranging from central carbon metabolism to DNA synthesis [1,2,3]. Thus, a sufficient iron pool needs to be maintained to sustain the production of iron-utilizing proteins. However, excess iron causes problems, especially in an oxidative atmosphere. During aerobic respiration, H_2_O_2_ was produced as a byproduct [4]. It can react with ferric iron (Fe^2+^) via the Fenton reaction [5,6] and produce reactive oxygen species (ROS). ROS can cause damage to proteins and DNA, resulting in cell death eventually [5,7,8,9]. Therefore, iron metabolism should be tightly controlled in order to maintain a sufficient iron pool to produce iron-utilizing proteins but to avoid excess iron that causes cell damage via ROS.

In *Escherichia coli* and in most Gram-negative bacteria, iron homeostasis is maintained by the ferric uptake regulator (Fur) together with the Fe^2+^ cofactor, which precisely controls the expression of genes functioning in iron metabolism [1,2,3]. Interestingly, even Fur without an iron cofactor (*apo*-Fur) can act as regulator, both in non-pathogenic *E. coli* [2] and some pathogenic bacteria [10,11,12]. Fur is able to repress and activate the expression of genes. Fur binds to Fe^2+^ and represses the transcription of iron-uptake genes by binding directly to their −35 and −10 promoter regions, which blocks the access of RNA polymerase [13,14,15]. Fur activated the expression of iron-storage genes [16] and several genes functioning in various cellular processes [2,17]. Fur activation is mediated by binding to extended sites between 60 and 240 bp upstream of the transcriptional start site (TSS). Gene expression is exerted either by stimulation of transcription by recruitment of RNA polymerase or by transcriptional silencing via the histone-like nucleoid-associated protein, H-NS [16]. Additionally, Fur also activates gene expression indirectly via regulation of the small RNA RyhB [3,18,19]. 

In *E. coli*, the Fur regulon has been extensively investigated via both in vitro DNA-binding [20,21] and in vivo ChIP-seq [2,3] experiments. Altogether, the current potential Fur regulon contains approximately 69 transcriptional units [2,3,22,23]. Beyond controlling iron metabolism, Fur controls the expression of several genes functioning in DNA synthesis, energy metabolism, and biofilm development [2]. However, given that several hundreds of genes have been identified to respond to Fur and iron in global assays [2], putative genes in Fur regulon could be hidden due to the limitations of experimental conditions in various laboratories. The present understanding of iron and Fur regulation could be limited. Additionally, is Fur the sole transcriptional factor responding to iron in *E. coli*? The answer to this question still seems to be obscure. A better understanding of iron and Fur regulation is required. 

In the present work, by integrating a high-resolution transcriptomic study of the Fur wild-type and knockout *E. coli* K-12 strains in the presence or absence of iron with high-throughput ChIP-seq assay and physiological studies, we re-studied the regulatory roles of iron and Fur systematically. Firstly, we proved that the transcriptional response of *E. coli* to iron was mediated solely by Fur. Next, we not only expanded the Fur regulon but also ascertained that a significant difference existed between the regulations of Fur on the two groups of genes under its direct repression and activation. Finally, we found that Fur linked iron metabolism to more fundamental processes than previously reported, and Fur exhibited systemic regulations on carbon metabolism, respiration, and motility. As a result, we interpreted the regulatory roles of iron and Fur regulation more comprehensively.

## 2. Results

### 2.1. Fur Is the Sole Regulatory Factor Responding to Iron

It is well known that iron metabolism is controlled mainly by Fur in *E. coli*. However, it is obscure whether Fur is the sole regulatory factor responding to iron or not. To answer this question, we studied the transcriptomes of the Fur wild-type and knockout strains grown in minimal medium with or without supplementation of iron via high-resolution RNAseq, which we had applied to extend the regulons of several global regulators including Crp and Cra [24,25]. As listed in Figure 1A, four pairs of transcriptome comparisons could be performed. We identified the differentially expressed genes (DEGs) in each of the four comparisons (Figure 1B–F). For the Fur wild-type strain of NCM3722, 58 DEGs were identified by comparing transcriptomes obtained under growth conditions with or without iron, and the bulk of the DEGs were repressed by iron, indicating that the signal of iron inhibited gene expression primarily in the wild-type strain (Figure 1B,C, Appendix A). In contrast, for the Fur-knockout strain of CY405, no DEGs were identified when comparing the transcriptomes obtained in equal conditions (Figure 1B,D). Thus, *E. coli* did not transcriptionally respond to iron in the absence of Fur. These observations showed clearly that Fur was the sole regulatory factor controlling the iron response, at least in the conditions that we studied.

Since iron is the cofactor of Fur and Fur is highly functional with iron, as expected, more DEGs were identified in the comparison of the Fur wild-type strain versus the Fur-knockout strain with iron (Figure 1B,E, Appendix A) than in the similar comparison of the two strains without iron (Figure 1B,F, Appendix A). Combining the two comparisons, 560 DEGs were identified. Accordingly, these DEGs were subjected to a direct or an indirect regulation of Fur.

### 2.2. Fur Regulon Is Expanded by Combined Analysis of ChIP-seq and RNA-seq

In order to identify the detected DEGs mentioned above that were regulated directly by Fur, we applied a ChIP-seq assay to study the binding sites of Fur in the strain grown in the same medium as that used in the RNA-seq assay. By combining the two ChIP-seq results obtained in the growth conditions with or without iron, 596 binding sites of Fur on the chromosome were identified (Figure 2A, Appendix A). Several examples of the Fur-binding peaks were shown in Figure 2B. We noted that the bigger binding peak carried a higher value of the fold enrichment of the peak. In total, there were 377 peaks located in the promoter region and 219 peaks located in the CDS region (Figure 2C, Appendix A). 

ChIP [26] and ChIP-seq [3,27,28] have been widely used to measure the strength of transcriptional factor (TF) binding to specific target sites. Hence, by using the parameter of fold enrichment of the peak as a proxy of the binding strength of Fur, we identified that the binding strength of Fur was much stronger in the promoter region than that in the CDS region (Figure 2D), supporting its role acting as a transcriptional factor. Within the 377 binding sites located in the promoter region, there were 24 sites located in intragenic regions of divergent promoters. Thus, 401 transcriptional units could be controlled by Fur directly.

We then integrated the RNA-seq data with the ChIP-seq results to characterize the genes under the putative direct control of Fur. We listed the following selection criteria. If the expression of the first gene in each of the 401 transcriptional units was identified as DEG which was responding to Fur in the RNA-seq comparison of the Fur wild-type strain versus the Fur-knockout strain with (Figure 1E) or without iron (Figure 1F), this transcriptional unit was characterized as being under the direct control of Fur. Accordingly, 84 and 28 transcriptional units were respectively identified in the growth conditions with or without iron supplementation (Figure 2E), with 23 units identified in both conditions. In total, we found that 89 transcriptional units were controlled by Fur directly, with 39 units being identified for the first time and with the remaining 50 units being consistent with the reported studies (Figure 2F, Appendix A). These transcriptional units contained 123 DEGs, accounting for approximately 22% of all the DEGs identified (Appendix A).

Figure 3A illustrated several exampled binding peaks identified in this work, including Fur regulon genes that were newly identified and reported. The fold enrichment of these peaks ranged from 1.15 to 22.39, covering most of the peaks carrying weak to strong binding signals. To further substantiate the findings of our ChIP-seq, several of the Fur bindings to identify promoters, especially those with weak binding peaks, were further validated via in vitro electrophoretic mobility shift assay (EMSA) (Figure 3B). Thus, we expanded the Fur regulon greatly. 

Since we identified that a mere twenty-two percent of DEGs were subject to a direct control of Fur, a large portion of these DEGs were controlled by Fur in an indirect manner. We looked into the contribution of the small RNA of RyhB by comparing our results (Appendix A) to the RyhB reglon [3] that was defined recently. In total, we found that 71 DEGs were also controlled by RyhB, with 26 DEGs subject to a dual control of both RyhB and Fur and leaving 45 DEGs controlled by RyhB exclusively. Hence, less than ten percent of all the DEGs identified in this work were controlled by Fur indirectly via RyhB. As a result, other factors would contribute to the indirect regulation of Fur, and we proposed that the effect of ROS [5,6] could be considered.

### 2.3. Discrepancies Existed between the Regulations of Fur on the Genes under Its Direct Repression and Activation

In the 89 transcriptional units identified, 22 units were activated by Fur, and 67 units were repressed by Fur (Figure 4A), which supported the primary repressive role of Fur on gene expression [2,3,22,23]. We next investigated the possible difference between the regulations of Fur on these two groups of genes under its repression and activation.

High ChIP-seq signal of a TF usually corresponds to strong binding strength of this TF to the targets [3,27,28]. Accordingly, we compared the binding strength of Fur based on the parameter of the fold enrichment of the peak on these two groups of genes and identified that the binding strength of Fur on the genes repressed by Fur was significantly stronger than that on the genes activated by Fur (Figure 4B). This finding suggested that these two groups of genes could contain a distinct consensus sequence of Fur-binding motifs. We next performed MEME motif [29] analysis of the Fur-binding motifs of the genes in the two groups and found that their consensus sequences were quite different indeed (Figure 4C,D). The consensus sequence of Fur-binding motif obtained by the genes that were repressed by Fur in this work was a reverse complementary sequence that was AT-rich (Figure 4C), supporting the one reported [2]. The consensus sequence of the Fur-binding motif obtained by the genes that were activated by Fur was also AT-rich, but it was much shorter and was not a reverse complementary sequence (Figure 4D). Based on the identified consensus sequence, the binding site of Fur in each promoter was predicated (Appendix A). The position of each of the binding sites relative to the TSS is shown in a box diagram (Figure 4E). The binding sites of Fur on the genes under its repression were near to the −35 and −10 regions of the promoters, with a median at −20 bp. In contrast, Fur-binding sites on the genes under its activation were far away from the −35 and −10 regions, with a median at −74 bp. These findings supported reported mechanisms of Fur repression and activation. 

The difference in the binding site of the two groups of genes could affect their responses to iron or Fur. Accordingly, we compared the fold changes in the shared DEGs that were simultaneously identified in the two transcriptome comparisons of the Fur-knockout strain versus the Fur wild-type strain grown in the conditions with and without iron (Figure 1E,F, Appendix A). Furthermore, these DEGs were controlled by Fur directly. We observed that the genes activated by Fur showed a similar fold of changes between the conditions with or without iron, whereas genes repressed by Fur showed a much bigger fold of changes when iron was supplied (Figure 4F), indicating that genes repressed by Fur were more sensitive to iron. In this line, we quantified the mean of the fold changes obtained in genes repressed and activated by Fur directly in the transcriptome comparison of the Fur wild-type strain versus the Fur-knockout strain, respectively. As compared to Fur-activated genes, Fur-repressed genes showed statistically bigger fold changes (Figure 4G, Appendix A), indicating a relatively greater regulatory role of Fur on the genes that were inhibited by Fur. 

We also conducted a correlation analysis between the degree of Fur regulation and the binding strength of Fur on the Fur regulon genes under its activation and repression together (Figure 4H). Interestingly, we found a weak positive correlation, which was consistent with the reported findings of other TFs [30,31]. Altogether, the response of the gene to Fur showed such a general trend that the greater the degree of regulation, the stronger the binding ability of Fur to this gene, and vice versa. 

Therefore, we found that Fur-repressed genes were relatively more sensitive to iron or Fur regulation than Fur-activated genes, and this discrepancy could have resulted from the difference in the Fur-binding sites of the two groups of genes.

### 2.4. The Regulatory Role of Fur Is Far beyond Maintaining Iron Homeostasis

Since we identified quite a large number of transcription units under the direct control of Fur, we could decipher the regulatory function of Fur more comprehensively. Accordingly, the Fur-controlled transcriptional units identified in this work (Appendix A) were classified into functional groups according to the GO term annotation of the first gene in each unit. Interestingly, both Fur-activated and Fur-repressed transcriptional units could be organized into a consistent fashion of various categories (Figure 5). The functional group of ‘Iron homeostasis’ contained the largest number of transcriptional units, accounting for thirty percent of all the transcriptional units identified. The remaining seventy percent functioned in various fundamental processes, including carbon metabolism, respiration, and motility. Thus, consistent with the literature [2,3], the regulatory function of Fur was beyond regulating iron homeostasis. However, the regulatory roles of Fur were elaborated further in this work.

### 2.5. Fur Coordinates Iron and Carbon Metabolism

We noted that many genes functioning in carbon metabolism responded to iron and Fur directly or indirectly. Five of the six Fur regulon genes (*mdh*, *mqo*, *gpmA*, *sdhA,* and *sdhB*) identified here functioning in carbon metabolism individually participated in the TCA cycle and the glycolysis (Figure 6A, Appendix A). The direct regulation of Fur on *mdh* and *mqo* was newly identified in this work, and its regulation on the other three genes was consistent with that reported in the literature [2,3,21]. Interestingly, more DEGs under the indirect control of Fur were also identified, and these genes functioned in various carbon-metabolism-related pathways, including glycolysis, the TCA cycle, the glyoxylate shunt, and the pentose phosphate pathway (Figure 6A). 

We mapped these genes together with the genes functioning in enterobactin biosynthesis to their metabolic pathways (Figure 6B). As shown in the map, enterobactin biosynthesis tightly connected the upper part of glycolysis, the pentose phosphate pathway, and the pathway of the chorismate biosynthesis. Interestingly, DEGs involved in these processes all negatively responded to Fur directly or indirectly. Since enterobactin biosynthesis needed the carbon skeletons generated in these carbon metabolism pathways, it was reasonable that these genes responded to Fur in an orchestral way. The mechanisms of the genes under the indirect regulation of Fur still needed to be discovered. 

We also observed asynchronous responses of DEGs identified in the TCA cycle to Fur regulation (Figure 6B). It seemed that when iron was deficient, Fur was inactivated, and the increased expression of *mqo*, *mdh*, and *gltA* possibly resulted in the high flux of citrate generation, whereas the decreased expression of other genes functioning downstream of citrate in the TCA cycle possibly caused the further accumulation of citrate. Given that citrate was a putative iron-chelating agent [32,33], citrate accumulation would also help the bacteria to alleviate the iron deficiency condition.

### 2.6. Fur Activates Anaerobic Respiration but Represses Aerobic Respiration

We found that two genes related to respiration could be controlled by Fur directly (Figure 7A, Appendix A). One was *hybO*, which was also reported as a Fur regulon gene [2,3]. The other one was *dmsB*, which was newly identified in this work. Although *dmsB* was within the *dmsABC* operon, which was under the indirect regulation of Fur via RyhB, given the strong binding peak of Fur identified in the upstream of *dmsB* (Appendix A), *dmsB* could have its individual promoter and be controlled by Fur directly. Further proofs were required to support this point. 

We also identified that more than 20 genes functioning in respiration were differentially expressed in the conditions of Fur wild-type versus Fur knockout (Figure 7A), including the entire operons of *nuoABCEFGHIJKLMN* encoding NDH-1, *frdABCD* encoding fumarate reductase, and *cyoABCDE* encoding ubiquinol oxidase. Fur was reported to bind to the promoter region of *cyoA* [20]. However, this binding peak was not identified in this study, possibly because of the uncontrolled variation in experiments. As a supplement, we applied EMSA to study the possible binding of Fur to some of the promoters (Figure 7B). We observed that Fur respectively bound to the promoters of *nuoA* and *frdA*, suggesting a direct control of Fur over these promoters. Additionally, as identified here (Appendix A) and as previously reported [3], *narZY* encoding nitrate reductase participating in anaerobic respiration belongs to the *narUZY* operon and was under the direct control of Fur. Therefore, the bulk of these differentially expressed genes functioning in respiration was controlled by Fur directly. 

We mapped these genes in the electron transfer chains, and an interesting phenomenon appeared (Figure 7C,D). We found that Fur activated the expression of genes functioning in anaerobic respiration (Figure 7C) but repressed the expression of genes functioning in aerobic respiration (Figure 7D). Therefore, Fur could take an important role in the regulation of respiration to coordinate iron metabolism. 

### 2.7. Fur Activates the Expression of Motility Genes

The final group of genes that attracted our eyes was mobility-related genes. More than 40 motility genes in the entire hierarchy of flagellar cascade (class I, class II, and class III) [34] responded to Fur (Figure 8A), accounting for nearly 10% of all the DEGs identified (Figure 1E,F). Although we could find the most systematic response of the motility genes to Fur regulation, this finding was not surprising because Fur has been identified to bind to the promoter region of *flhDC* in *E. coli* K-12 strains [20], and the genes of the *flhDC* operon encode the master activator of flagellum biosynthesis in *E. coli*. Thus, Fur could regulate the expression of the motility genes indirectly through its direct control of the *flhDC* operon. Indeed, we also observed a visible binding peak of Fur in the promoter region of *flhDC* (Appendix A). However, this binding peak was not identified as significant, possibly because this weak binding peak was masked by other strong binding peaks in our ChIP-seq analysis. 

Interestingly, we noted that there were discrepancies in the reported regulatory mode of Fur on motility genes in various *E. coli* strains. In a uropathogenic *E. coli* strain, Fur repressed its motility via its direct regulation of flhD [35]. However, in another avian pathogenic *E. coli* strain, Fur activated its motility [36]. Here, we found that the motility genes positively responded to Fur (Figure 8A and Appendix A), whereas the documented regulatory mode of Fur on *flhDC* in *E. coli* K-12 strains was repression [22]. To substantiate this finding, we studied the motility of both the Fur-knockout and wild-type strains. Indeed, the swimming ability of the Fur-knockout strain was largely impaired, not only on the soft agar plates (Figure 8B,C) but also in the liquid medium (Figure 8D,E). In parallel, no flagella were observed in the Fur-knockout strain (Figure 8F). This evidence supported that Fur activated the flagellum biosynthesis by activating the expression of *flhDC* when iron was replete in the *E. coli* K-12 strain of NCM3722 that used in this study. 

Given the discrepancies in the reported regulatory mode of Fur on motility genes in pathogenic *E. coli* strains, in order to further clarify the regulatory effect of Fur on motility in *E. coli* K-12 strains, we studied the possible regulation of Fur on motility in another widely used *E. coli* K12 strain, MG1655. To our surprise, we found that Fur showed no significant regulation on its motility (Appendix A), neither on the expression of *flhDC* nor on other flagella genes (Appendix A). We noted that the promoter sequences of *flhDC* covering the binding site of Fur in both strains of NCM3722 and MG1655 were identical, but the transposons of insertion sequences inserted upstream of the promoter of *flhDC* in MG1655 and NCM3722 were different [37] (Appendix A). The varied insertion sequences could account for the difference of the Fur regulation in the two strains. As reported and also identified in this work, the IS1 insertion in MG1655 resulted in the high expression of FlhDC and a hyper-motile phenotype [38,39,40] (Appendix A). Given the highly expressed basal level of FlhDC, it was not surprising that the regulatory effect of Fur on *flhDC* was not significant in MG1655. However, in the case of NCM3722 derivatives, the expression level of FlhDC would be very low since its motile speed of the fur-knockout strain (Figure 8B) on the soft agar plates (0.11 mm/h) was much lower than that of the fur-knockout strain derived from MG1655 (2.43 mm/h) (Appendix A). Consequently, the positive regulation of Fur on FlhDC appeared to be significant in the NCM3722 background since its basal expression level of *flhDC* was low. Therefore, this evidence suggested that the regulatory effect of Fur on the motility of *E. coli* strains was unique to a particular strain, which needed to be studied in a case-by-case context.

Additionally, since our ChIP-seq identified that Fur bound to the promoter regions of two motility-related genes (*ycgR* and *flgL*) (Appendix A), and Fur was also reported to bind to the promoter region of *fliE*, *fliF,* and *fliM* [3], Fur could also directly control the expression of some of the motility genes.

## 3. Discussion

In this work, we re-investigated the regulatory roles of iron and Fur in an *E. coli* K-12 strain systematically. In this process, we discovered several intriguing features of Fur regulation. 

We extended the Fur regulon greatly with the newly identified Fur-controlled transcriptional units (Figure 2F, Appendix A). Note that we have identified several hundreds of Fur-binding sites in the promoter regions (Figure 2A,C). However, consistent with previous findings [3], many of the downstream genes controlled by these Fur-bound promoters did not respond to Fur. Given the fact that many promoters are under coordinated control of various regulators [25,41], it was very possible that the regulations of Fur could be hampered by other regulators, and Fur regulation would emerge in some specific conditions. As a result, these promoters bound by Fur could control putative Fur regulon genes, and we will not be surprised if more Fur regulon genes are validated in the future.

We noted that genes responding to Fur (Figure 1) and Fur-binding peaks were still detected, even in the condition of no iron supplied in the growth medium (Figure 2). This finding seems to suggest that iron depletion is not sufficient to fully abrogate the iron-dependent activity of Fur and/or that several genes are regulated by *apo*-Fur [2]. However, we are unable to ensure that the medium without iron supplied rendered an absolutely iron-depleted condition since traces of iron contained in other chemical components of the medium were unavoidable. 

Intriguingly, we ascertained that there were significant discrepancies between the regulations of Fur on the genes under its repression and activation. These discrepancies were identified in the aspects of the Fur-binding strength (Figure 4B), the consensus sequence of Fur-binding sites (Figure 4C,D), and the sensitivity of genes to iron or Fur regulation (Figure 4F,G). This finding could explain why, in many cases, Fur was easily identified as a repressor given the overall lower sensitivity of Fur-activated genes to iron and Fur regulation as compared to genes under its repression. 

With the newly extended Fur regulon, we could better understand the regulatory roles of Fur comprehensively. Besides its well-known roles in maintaining iron metabolism, iron or Fur controlled more numbers of fundamental processes (Figure 5) than reported [2,3]. Our interests were focused on the systemic regulation of Fur on genes functioning in carbon metabolism, respiration, and motility.

We identified that Fur coordinated iron and carbon metabolism. Since only a small number of genes functioning in carbon metabolism were identified to be controlled by Fur directly, including *gpmA* [2,21] in glycolysis and *sdhABCD*-*sucABCD* [3] in the TCA cycle, the joint regulation of Fur on iron and carbon metabolism was not studied systematically. In this work, owing to the high quality of the RNA-seq and ChIP-seq data obtained, we found that many genes functioning in carbon metabolism pathways responded to Fur directly or indirectly (Figure 6A). By mapping all of these genes instead of only genes that were directly controlled by Fur to the pathways together (Figure 6B), we were able to uncover the possible reasons for the close link between carbon and iron metabolism in a new dimension. Given the orchestral responses of the enterobactin biosynthesis genes and these carbon metabolism genes, one possible reason for this connection was that Fur coordinated iron and carbon metabolism to synthesize enterobactin in order to assimilate iron. Given the asynchronous responses of genes in the TCA cycle to Fur regulation (Figure 6B), a final accumulation of citrate, a putative iron-chelating agent [32,33], would be expected when iron was deficient and Fur was inactivated. As a result, Fur appeared to control carbon metabolism to help the cells to assimilate iron when iron was absent. However, the molecular mechanisms of the genes in these processes under the indirect regulation of Fur need to be explored. Moreover, the small RNA of RyhB could be a first candidate to study [3,42,43].

We identified that Fur would be an important regulator controlling respiration. Interestingly, Fur inhibited genes involved in aerobic respiration but activated genes involved in anaerobic respiration, with the bulk of these genes under the direct control of Fur (Figure 7). To our knowledge, few reports have identified the direct regulatory effect of Fur on both anaerobic and aerobic respiration genes in one study, which could be the possible reasons why the regulation of Fur on respiration was overlooked previously. Why would Fur prefer anaerobic respiration? We note that ROS are generated as by-products of aerobic respiration [4,9], and ROS will oxidize the Fur cofactor of Fe^2+^ into Fe^3+^, resulting in the inactivation of Fur. This predication is further supported by the observation that a lower amount of Fe^2+^ was detected during aerobic growth as compared to that in anaerobic growth [44]. Thus, Fur would benefit itself by inhibiting aerobic respiration but activating anaerobic respiration to maintain its own activity. 

Our evidence (Figure 8, Appendix A) indicated a close link between iron metabolism and bacterial motility. However, given the inconsistent results that we obtained on the two non-pathogenic *E. coli* strains (Figure 8, Appendix A) and the discrepancies in the regulatory mode of Fur on motility genes in pathogenic bacteria reported [35,36,45], we suggested that the regulation of Fur on bacteria motility was unique to a particular strain. The proposed reason could be that the promoter region of *flhDC* is a hot spot for transposon insertion [34,38], and the inserted transposon could complicate Fur regulation on this promoter in various genetic backgrounds accordingly. Moreover, motility affected the virulence of pathogenic bacteria [46,47]. Thus, Fur could be a putative target in drug design, but this strategy needs to be considered in a manner personalized to a unique pathogen. 

In summary, we deciphered iron and Fur regulation in *E. coli* comprehensively in this work. These findings help us to better understand the regulatory mechanisms of the physiology of *E. coli* that is applied systemically when it faces variation in iron content.

## 4. Materials and Methods

### 4.1. Construction of Bacterial Strains

The strains that were used in this study are listed in Appendix A. Derivatives of the wild-type *E. coli* K-12 strains NCM3722 [48] and MG1655 were all constructed by the *λ*-Red system [49]. Strain NCM3722 and its *fur*-deficient derivative strain CY405 were used in the transcriptome study. Strain CY1099 was the *fur*-6×his tagged derivative of NCM3722 and was used in the ChIP-seq study. Since NCM3722 was non-motile due to the mutation in the flagella structural protein FliC [37], the *fliC* fixed wild-type strain CY508 [34] and its *fur*-deficient derivative CY856 were used in the motility assay. Similarly, MG1655 and its *fur*-deficient derivative CY856 were also used in the motility assay. 

### 4.2. RNA-seq Assay and Analysis of Transcriptome Data

Strains were grown in the N^-^C^-^ minimal medium [48] with 20 mM NH_4_Cl as the nitrogen source and 20 mM sorbitol as the carbon source. Since we had previously identified that Fur regulon genes strongly responded to sorbitol [25], sorbitol was applied as the sole carbon source in order to better characterize the Fur regulon. When necessary, 1 mg/L ammonium iron citrate was supplied in the medium. All growths were carried out in three steps in a 37 °C water bath shaker as described previously [48]. Samples were collected at mid-exponential phase with OD_600_ ~0.4 to extract total RNA. Total RNA extraction and RNA-seq assay were performed as described previously (Li et al., 2018) [24]. In brief, cells were quenched with pre-cooled quenching buffer (60% Methanol, 70 mM HEPES) and collected via centrifugation. Cell pellets were resuspended in 100 μL lysis buffer [10% glucose, 12.5 mM Tris (pH 7.6), 10 mM EDTA (pH 8.0), 200 U mL^−1^ RNase inhibitor (Applied Biosystems, Waltham, MA, USA) and 40 mg mL^−1^ lysozyme (Sangon, Shanghai, China)] and treated for 3–5 min at RT. Then, 1 mL RNAiso Plus (Takara, Dalian, China) was added, followed by twice extraction with chloroform. The supernatant was then treated in 80% ethanol and purified by the PureLink miRNA Isolation Kit (Invitrogen, Waltham, MA, USA). The RNA that was eluted was further treated with Turbo DNase (Ambion, Waltham, MA, USA), then quantified and qualified by the NanoDrop 2000 (Thermo, Waltham, MA, USA) and 2100 Bioanalyzer (Agilent, Santa Clara, CA, USA). Strand-specific RNA sequencing was then performed. Ribosomal RNA was removed with the Ribo-Zero rRNA Removal Kit (Bacteria; epicentre). Then, mRNA was sheared to short fragments by adding fragmentation buffer, and first-strand cDNA was synthesized with random hexamer and reverse transcriptase. The second-strand cDNA was synthesized by adding GEX second-strand buffer, dNTP mix, RNase H, and DNA polymerase I. The fragments of cDNA were then purified and followed by end-reparation by using T4 DNA polymerase and Klenow DNA polymerase. Fragments were adenylated at their 3′ ends and ligated with sequencing adapters by T4 DNA ligase. Second-strand cDNA was degraded by the UNG enzyme and purified. Templates of cDNA with adapters were next enriched by PCR amplification and purification. The library was sequenced by using Illumina HiSeq 2000, PE125. Raw reads were filtered, and adaptors were trimmed. Clean reads were mapped to the NCM3722 genome (NCBI accession number for chromosome: CP011495.1, for plasmid F: CP011496.1) by using Bowtie2. The RPKM (reads per kilo bases per million mapped reads) method [50] was used to calculate the expression of each gene. Three independent total RNA extractions and transcriptome analyses by RNA-seq were performed for each condition. DEGs between the compared conditions were characterized by DEseq2 [51] with an adjusted *p*-value (*q*-value) < 0.01.

### 4.3. ChIP-seq Assay and Analysis of ChIP-seq Data

ChIP-seq assays were performed according to the method described previously [52]. Briefly, CY1099 were grown in the same medium with or without ammonium iron citrate (1 mg/L) as that used in the RNA-seq assay. Then, 40 mL of culture in the mid-exponential phase was cross-linked with 1.1 mL formaldehyde (37%) for 20 min at room temperature and quenched by the addition of glycine at a final concentration of 137 mM. The samples were centrifuged and washed twice in cold PBS buffer. Chromatins were sonicated to obtain soluble sheared chromatin with an average DNA length of 200–500 bp. Then, 100 μL chromatin was precipitated by Anti-6×His tag antibody (Abcam ab9108) at 4 °C overnight, and 20 μL chromatin was stored at −20 °C to save as the control of input DNA. The next day, 30 μL of protein G beads was added, and the samples were further incubated for 3 h. The beads were next washed once with buffer containing 20 mM Tris/HCl (pH 8.1), 50 mM NaCl, 2 mM EDTA, 1% Triton X-100, and 0.1% SDS; twice with buffer containing 10 mM Tris/HCl (pH 8.1), 250 mM LiCl, 1 mM EDTA, 1% NP-40, and 1% deoxycholic acid; and twice with TE buffer (10 mM Tris-HCl at pH 7.5, 1 mM EDTA). Precipitated DNA was then eluted by 300 μL of elution buffer (100 mM NaHCO_3_, 1% SDS); then, the DNA was treated with RNase A (8 μg/mL) at 65 °C and with proteinase K (345 μg/mL) at 45 °C. Sequencing libraries were constructed following the protocol provided by the INEXTFLEX^®^ ChIP-Seq Library Prep Kit for Illumina^®^ Sequencing (NOVA-514120, Bioo Scientific, Austin, TX, USA) and sequenced on Illumina Xten with the PE 150 method. To analyze the sequencing data, Trimmomatic (version 0.38) was used to filter out low-quality reads [53]. Clean reads were mapped to the *E. coli* NCM3722 genome by Bwa (version 0.7.15) [54]. Samtools (version 1.3.1) was used to remove potential PCR duplicates [55]. MACS2 software (version 2.1.1.20160309) was used to call peaks by default parameters with *p*-value < 0.05 [56]. Input DNA was used as a control to model the background noise. The parameter of fold enrichment of the peak indicated the strength of the ChIP-seq signal [3,27,28]. When a binding peak was identified in the conditions both with or without iron supplied, the averaged value of fold enrichment of the peak was documented.

### 4.4. Overexpression and Purification of Fur

Procedures of overexpression and purification of His-tagged Fur were performed as described previously [25]. In brief, the strain of CY668 [25] for the overexpression of His-tagged Fur were cultured in LB broth with kanamycin at 37 °C to the log phase (OD_600_ ~ 0.6). Subsequently, with the addition of 1 mM IPTG, cells were kept growing at 28 °C overnight and harvested by centrifuge. Then, purification of His-tagged Fur protein was performed following the instruction of Capturem His-tagged Purification Miniprep Kit (Clontech, Mountain View, CA, USA). The concentration and purity of the protein were respectively determined via Bradford assay and Coomassie stains of SDS-PAGE gels. Proteins with purity greater than 95% were obtained for the following EMSA assay.

### 4.5. EMSA of Fur

The promoter region of each target gene was PCR amplified by using the primers listed in Appendix A and purified to serve as a probe in EMSA. The DNA-binding reaction was set up as follows: 0.25 pmol DNA, 0–72 pmol Fur proteins in binding buffer (20 mM BisTris/borate (pH 7.0), 40 mM KCl, 1 mM MgCl_2_, 200 μM CoCl_2_, and 5% glycerol). The reaction mix was incubated at 37 °C for 15 min and separated on a 5% native polyacrylamide gel at 220 V for 40 min. After electrophoresis, the gel was stained by Gelview (EP1502, Bioteke Corporation, Beijing, China) and visualized by Gel Doc XR+ (Bio-Rad, Hercules, CA, USA). At least two independent repeats were performed for each target. 

### 4.6. Real Time Quantitative PCR (RT-qPCR)

Strains were grown in LB medium to the mid-exponential phase, and the total RNA were extracted as described previously [25]. The cDNA templates were synthesized following the instruction of a PrimeScript RT reagent Kit with gDNA Eraser (Takara). RT-qPCR assays were performed on the qTOWER real-time PCR thermocycler (Analytik Jena, Jena, Germany). The primers that were used are listed in Appendix A. The expression level of *flhD*, *flhC*, *fliC*, *flgD*, *pdeH*, *fliA,* and *flgA* was individually normalized to the expression level of *polA,* which served as the reference gene. The reaction mixture (20 μL) consisted of 10 μL SYBR Premix Ex Taq II (Tli RNaseH Plus) (Takara), 2 μL template of 10-fold diluted cDNA, 7 μL H_2_O, and 0.5 μL (10 μM) of each primer. No template controls (NTC) were run in parallel, and all reactions were run in triplicate. The relative expression was analyzed by using the ΔΔCt quantification method by qPCRsoft v3.2, where CT refers to the cycle threshold. Three independent experiments were performed.

### 4.7. Swimming Motility Assay on Soft Agar Plate

Mid-exponential phase cultures of strains grown in LB (1.0 μL) were incubated at 37 °C on soft LB agar plates [0.25% agar (Sigma-Aldrich, St. Louis, MO, USA)], and the diameters of swimming halos were measured during the process of incubation. Two independent experiments were performed.

### 4.8. Time-Lapse Microscopy and Trajectory Analysis of Single-Cell Motility

Bacterial strains were grown to mid-exponential phase and diluted to an OD_600_ of approximately 0.002 with a fresh culture medium in a 6-well microtiter plate (Polystyrene, tissue culture treated, flat bottom; Jet Bio-Filtration Co., Ltd., Guangzhou, China). The movement of bacterial cells and cell trajectories were respectively captured and analyzed as reported previously [34]. Data of trajectories including path length (μm) and duration [second (s)] were exported, and the motility speed of a single cell was calculated as the path length divided by the duration. Two independent experiments were performed.

### 4.9. Flagella Observation under Transmission Electron Microscope

Strains were grown to mid-exponential phase in LB medium and harvested by centrifugation at 4500 rpm. Cell pellets were washed with sterile double-distilled water (DDW) and resuspended in DDW to an OD_600_ of 1.0. Then, the flagella of bacterial strains were stained and observed under transmission electron microscope HITACH HT7700, as was described in a previous report [34]. Two independent experiments were performed.

### 4.10. Statistical Analysis

Unless specified, the statistical analysis applied in this work was performed by using the statistical tools in SigmaPlot software (version 12.5).

## Figures and Tables

**Figure 1 ijms-24-09078-f001:**
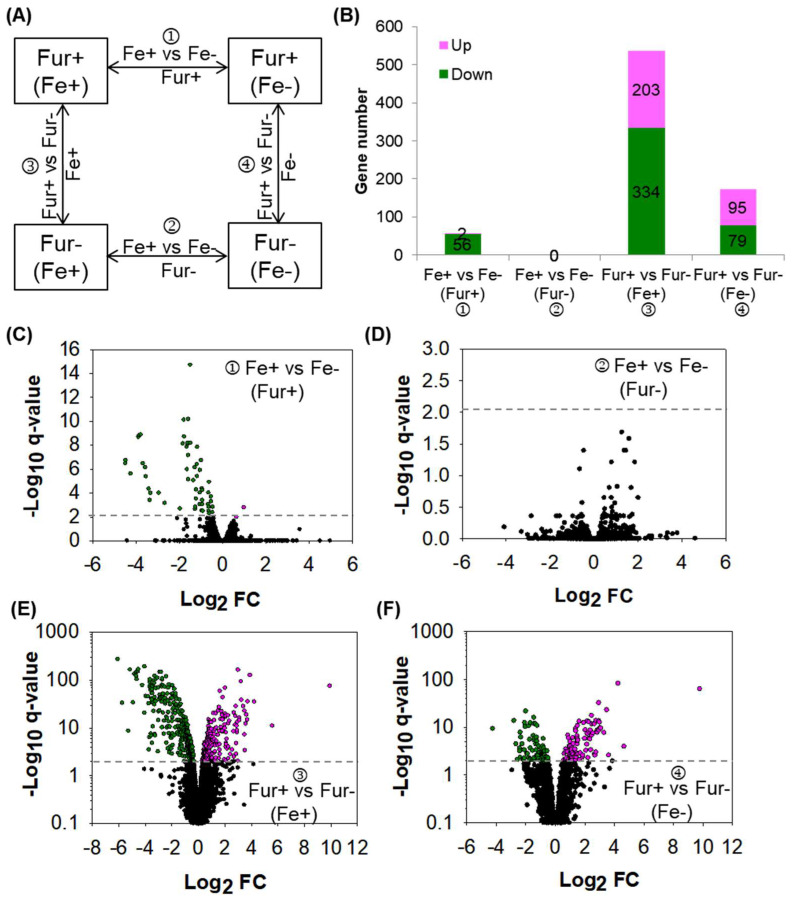
The DEGs identified between the Fur wild-type and knockout strains with or without iron supplied. (**A**) Experimental design of the four comparisons of transcriptomes. Fe+: with iron supplied in the medium; Fe−: without iron supplied in the medium; Fur+: Fur wild-type strain NCM3722; Fur−: Fur-knockout strain CY405. (**B**) The number of the DEGs identified in each of the four comparisons of transcriptomes. Up: up-expressed DEGs; Down: down-expressed DEGs. (**C**–**F**) Volcano map comparing the expression of genes in each of the four comparisons of transcriptomes. Horizontal lines point to *q*-value = 0.01. Pink dots indicate up-expressed DEGs. Green dots indicate down-expressed DEGs, and black dots indicate genes not differentially expressed (non-DEGs). (**C**,**D**) NCM3722 (**C**) or CY405 (**D**) grown in the medium with iron versus that without iron. (**E**,**F**) NCM3722 versus CY405 grown in the medium with (**E**) or without iron (**F**).

**Figure 2 ijms-24-09078-f002:**
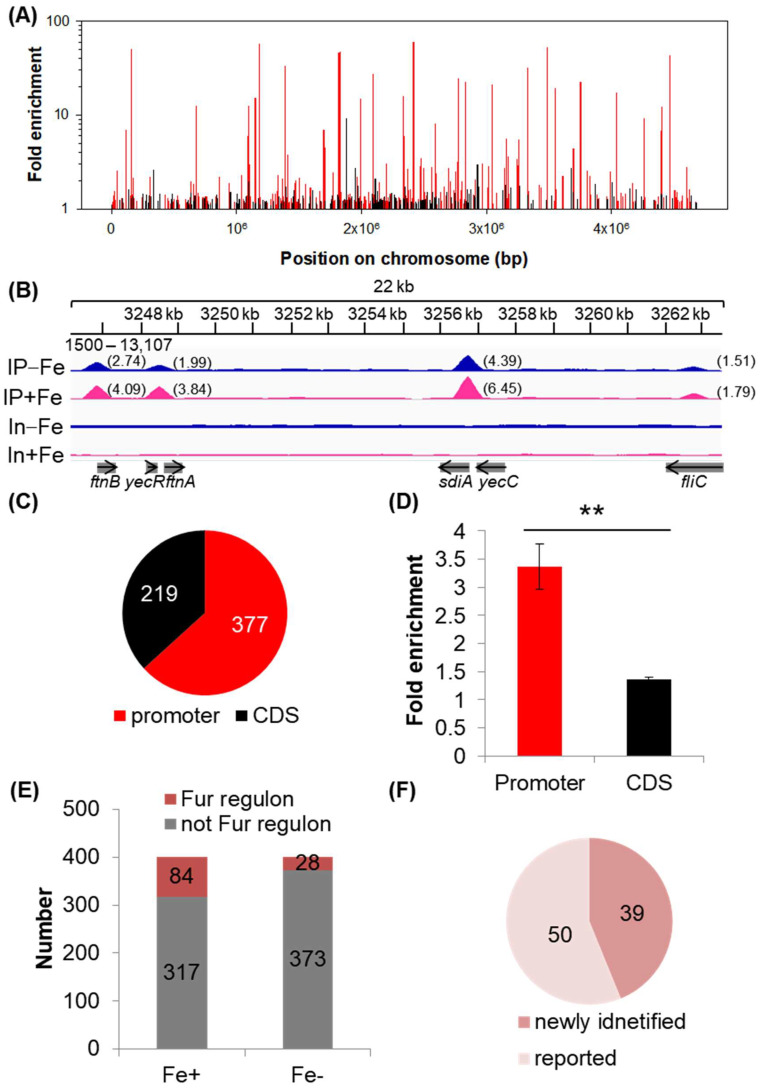
Identification of the Fur regulon by combined analysis of ChIP-seq and RNA-seq. (**A**) Distributions of the Fur-binding peaks identified in the genome. Red and black bars, respectively, represent peak signals identified in the promoter region (upstream of the start codon of a gene) and in the CDS region (coding sequence of a gene). The parameter of fold enrichment of the peak indicates the strength of the ChIP-seq signal. (**B**) Examples of enlarged peaks identified in the chromosome. This image shows a part of the Fur-binding profiles in the chromosome. IP−Fe/IP+Fe: ChIP-seq result in the growth condition without (IP−Fe) or with iron (IP+Fe); In−Fe/In+Fe: Input control result in the growth condition without (In−Fe) or with iron (In+Fe). ‘1500-13,107′ indicates the data range of the scale. Numbers in parentheses show values of fold enrichment of the peaks. (**C**) Peak numbers identified in the promoter region and in the CDS region. (**D**) Fold enrichment of the binding peaks. The fold enrichment of the binding peaks was expressed as the average ± SEM (*n* = 377 in promoter region; *n* = 219 in CDS region). ** indicates *p* ≤ 0.01 by Student’s *t*-test. (**E**) The number of the transcriptional units that were identified to be controlled by Fur directly out of the 401 transcriptional units bound by Fur. Fe+: with iron; Fe−: without iron. (**F**) The size of the Fur regulon identified combining the two conditions with or without iron.

**Figure 3 ijms-24-09078-f003:**
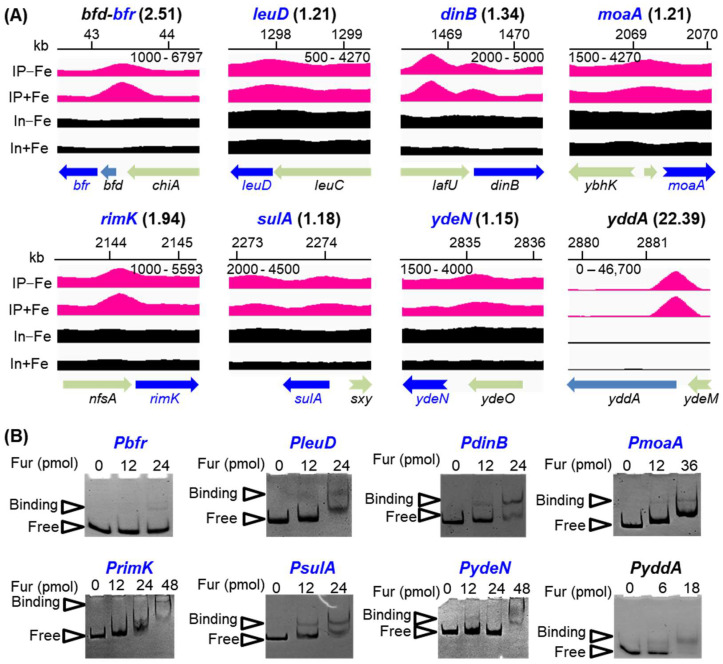
Binding peaks of Fur identified upstream of operons and validation of the peaks by EMSA. (**A**) IGV image of the Fur-binding peaks identified upstream of some operons on the chromosome. IP−Fe/In−Fe: ChIP-seq (IP−Fe) or input control (In−Fe) result in the growth condition without iron; IP+Fe/In+Fe: ChIP-seq (IP+Fe) or input control (In+Fe) result in the growth condition with iron. Numbers below the black line indicate the range of the peak scale. Arrow direction indicates gene direction, and arrow length indicates gene length. Swallowtail arrow means that only part of the gene is shown. Numbers in parentheses show values of fold enrichment of the peaks. (**B**) Validation of Fur-binding peaks via in vitro EMSA. Each promoter fragment was incubated with purified Fur protein, as described in the Materials and Methods. Free and binding probes were respectively indicated by triangles next to the image. In panels (**A**,**B**), Fur regulon genes newly identified in this work are shown in blue, and reported Fur regulon genes are shown in black.

**Figure 4 ijms-24-09078-f004:**
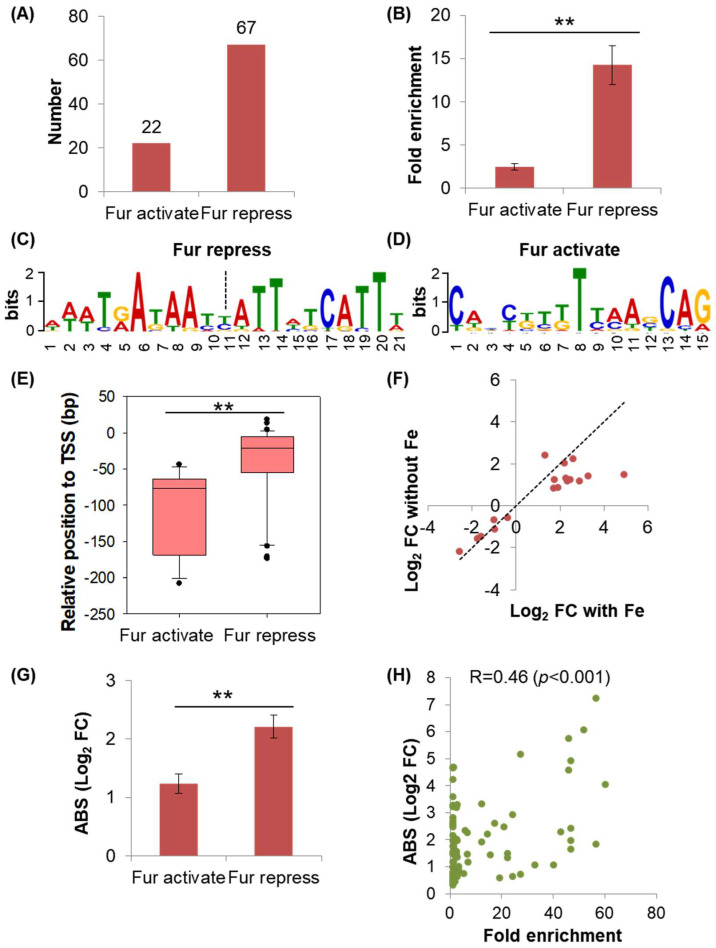
Characterization of the differences between the regulations of Fur on the genes under its repression and its activation. (**A**) Numbers of the transcriptional units directly repressed and activated by Fur. (**B**) Fold enrichment of the Fur-binding peaks identified in the transcriptional units repressed and activated by Fur. Similar to that shown in Figure 2D, the value in Appendix A was expressed as the average ± SEM with *n* = 22 (activated by Fur) and *n* = 67 (repressed by Fur). ** indicates *p* ≤ 0.01 by Student’s *t*-test. (**C**,**D**) Logo of the consensus sequence of the Fur-binding sites in the promoter regions of Fur-repressed (**C**) and -activated genes (**D**). The logo was analyzed by the MEME Suite (http://meme-suite.org/) with E-value = 1.3 × 10^−57^ in (**C**) and E-value = 0.01 in (**D**). The dashed line points to the center of the consensus sequence in (**C**). (**E**) Comparing the positions of Fur-binding sites relative to TSS in the genes under its repression and its activation. The central positions of Fur-binding sites relative to TSS were plotted in a box diagram. TSS means transcriptional start site of a gene. See the data in Appendix A. (**F**) Comparing the fold changes in DEGs identified in the conditions both without and with iron. The fold change in each gene obtained in the condition without iron was plotted against that with iron. The dashed line indicates y = x to guide the eye. See the data in Appendix A. (**G**) The absolute fold change in genes repressed and activated by Fur in the transcriptome comparison of NCM3722 versus CY105 in the growth condition with iron. The absolute value (ABS) of the fold change in the first gene in each directly Fur-controlled transcriptional unit was analyzed. The value shown was expressed as the average ± SEM with *n* = 20 (activated by Fur) and *n* = 64 (repressed by Fur). See the data in Supplemental Appendix A. (**H**) The correlation between the fold change in the gene and the strength of Fur binding to its promoter. The absolute values (ABS) of the fold changes in the first gene in each Fur directly activated and repressed transcriptional unit in (**G**) were plotted against their corresponding binding strength of Fur identified by ChIP-seq in Appendix A. The correlation coefficient R and *p*-value of the linear regression were analyzed via SigmaPlot. In panels (**B**,**E**,**G**), ** indicates *p* ≤ 0.01 by Student’s *t*-test.

**Figure 5 ijms-24-09078-f005:**
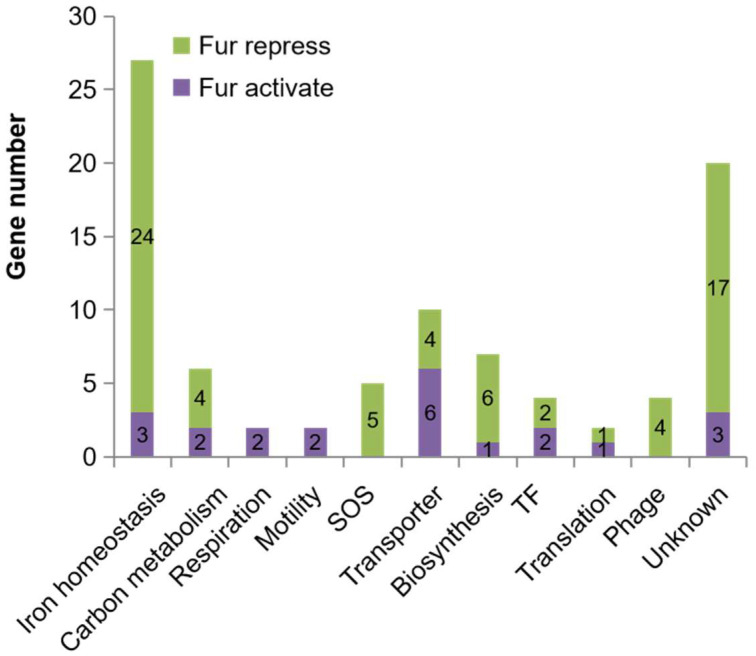
Gene numbers in each functional category. The 89 Fur-controlled transcriptional units in Appendix A were classified into functional groups according to the GO term annotation of the first gene in each unit. ‘Iron homeostasis’ group contains genes related to iron metabolism; ‘Carbon metabolism’ group contains genes functioning in central carbon metabolism and catabolism of various carbon substrates; ‘Respiration’ group contains genes encoding proteins functioning in aerobic or anaerobic respiration; ‘Motility’ group contains structural, biosynthetic, and regulatory genes of flagella, motility, and chemotaxis; ‘SOS response’ group contains genes belonging to the SOS system; ‘Transporter’ group contains genes encoding transporters of various substrates; ‘Biosynthesis’ group contains genes functioning in biosynthesis of various molecules; ‘TF’ group contains genes encoding transcriptional factors; ‘Translation’ group contains genes encoding proteins functioning in translation; ‘Phage’ group contains genes encoding prophage proteins; ‘Function unknown’ group contains genes with physiological functions un-annotated.

**Figure 6 ijms-24-09078-f006:**
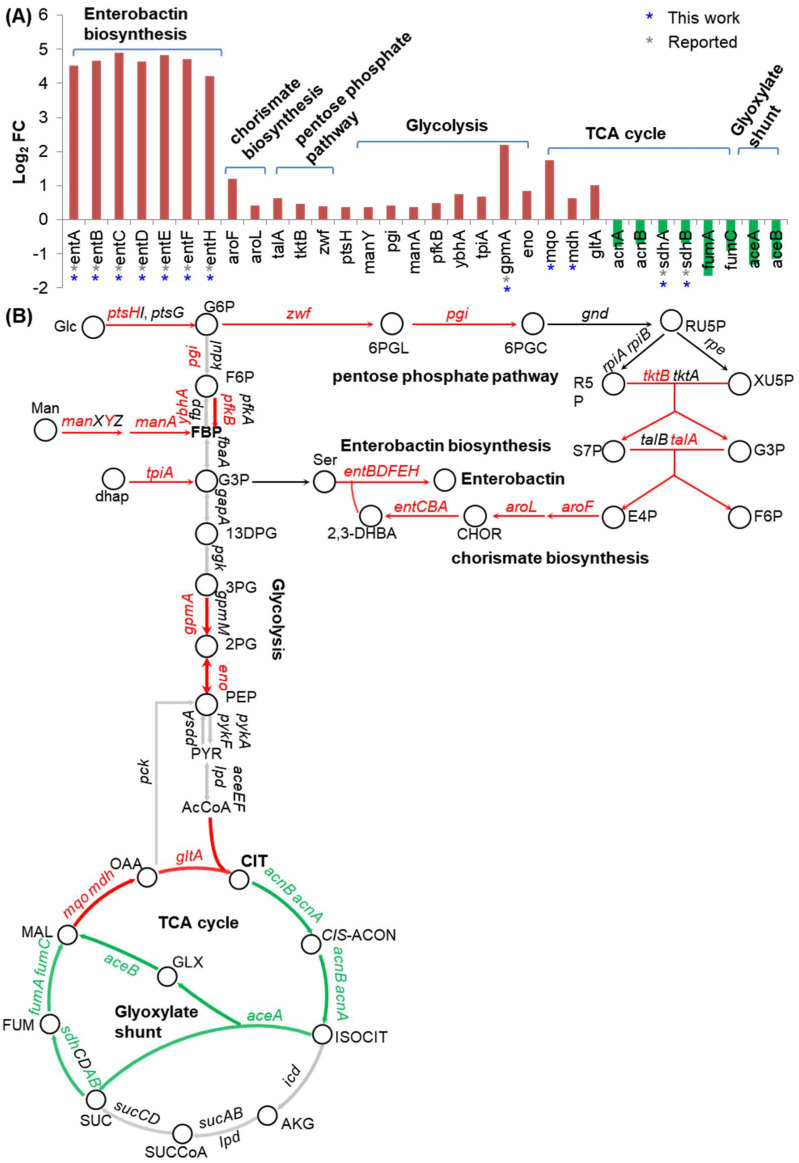
Fur-coordinated iron and carbon metabolism. (**A**) Fold changes in DEGs functioning in enterobactin biosynthesis and various forms of carbon metabolism in the transcriptome comparison of CY405 versus NCM3722. Green bars showed DEGs activated by Fur (directly or indirectly) and red bars showed DEGs repressed by Fur (directly or indirectly). DEGs that directly responded to Fur identified by ChIP-seq in this work were marked by stars in dark blue, and those that were identified in the literature were marked by stars in gray. (**B**) DEGs functioning in enterobactin biosynthesis and various pathways of carbon metabolism were mapped to the pathways. DEGs repressed by Fur (directly or indirectly) are shown in red, and DEGs activated by Fur (directly or indirectly) are shown in green. Non-DEGs are shown in black.

**Figure 7 ijms-24-09078-f007:**
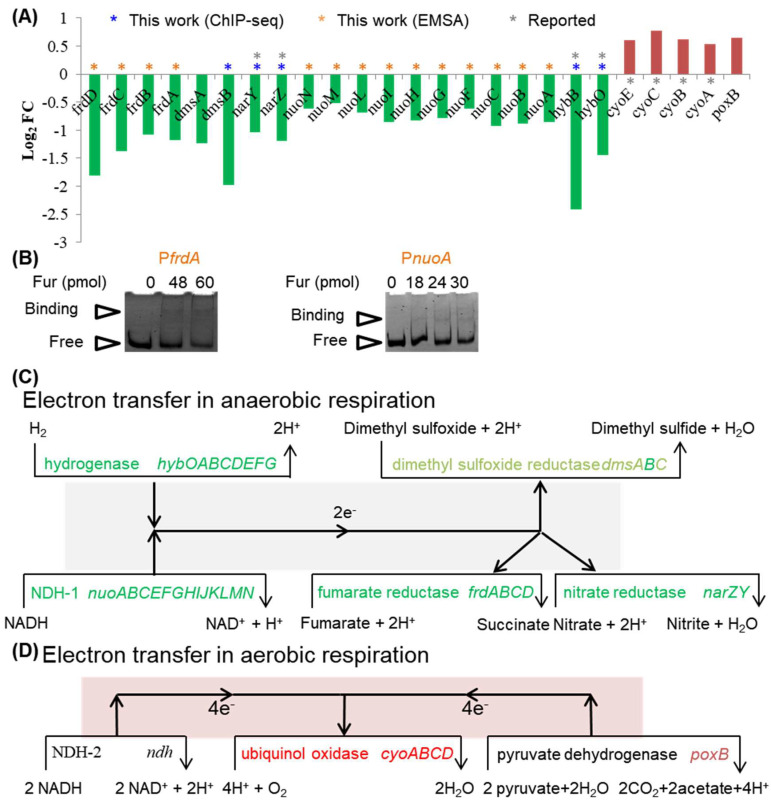
DEGs functioning in electron transfer of aerobic and anaerobic respiration. (**A**) Fold changes in DEGs functioning in aerobic and anaerobic respirations in the transcriptome comparison of CY405 versus NCM3722. Green bars showed DEGs activated by Fur (directly or indirectly), and red bars showed DEGs repressed by Fur (directly or indirectly). DEGs that directly responded to Fur identified by ChIP-seq in this work were marked by stars in dark blue, those identified by EMSA in panel (**B**) were marked by stars in orange, and those identified in the literature were marked by stars in gray. (**B**) Identification of Fur-binding promoters by EMSA. The promoter fragment of *frdA* (P*frdA*) or *nuoA* (P*nuoA*) was incubated with a purified Fur protein, as described in the Materials and Methods. Free and binding probes were respectively indicated by triangles next to the image. At least two independent repeats were performed for each assay, and one representative image is shown. (**C**) DEGs functioning in anaerobic respiration were mapped to the electron transfer chains. Electrons could be transferred from NADH to one of the three substrates (fumarate, nitrate, and dimethyl sulfoxide). The enzymes that functioned in this process included NDH-1, fumarate reductase, nitrate reductase, and sulfoxide reductase. Alternatively, electrons could be transferred from H_2_ to fumarate or dimethyl sulfoxide. The enzymes that functioned in this process included hydrogenase, fumarate reductase, and sulfoxide reductase. (**D**) The DEGs functioning in aerobic respiration were mapped to the electron transfer chains. Electrons could be transferred from NADH to O_2_ or from pyruvate to O_2_. The enzymes that functioned in this process included NDH-2, pyruvate dehydrogenase, and cytochrome o ubiquinol oxidase. In panels (**C**,**D**), DEGs repressed by Fur are shown in red (directly) and light red (indirectly). DEGs activated by Fur are shown in green (directly) or olive green (indirectly). Non-DEGs are shown in black.

**Figure 8 ijms-24-09078-f008:**
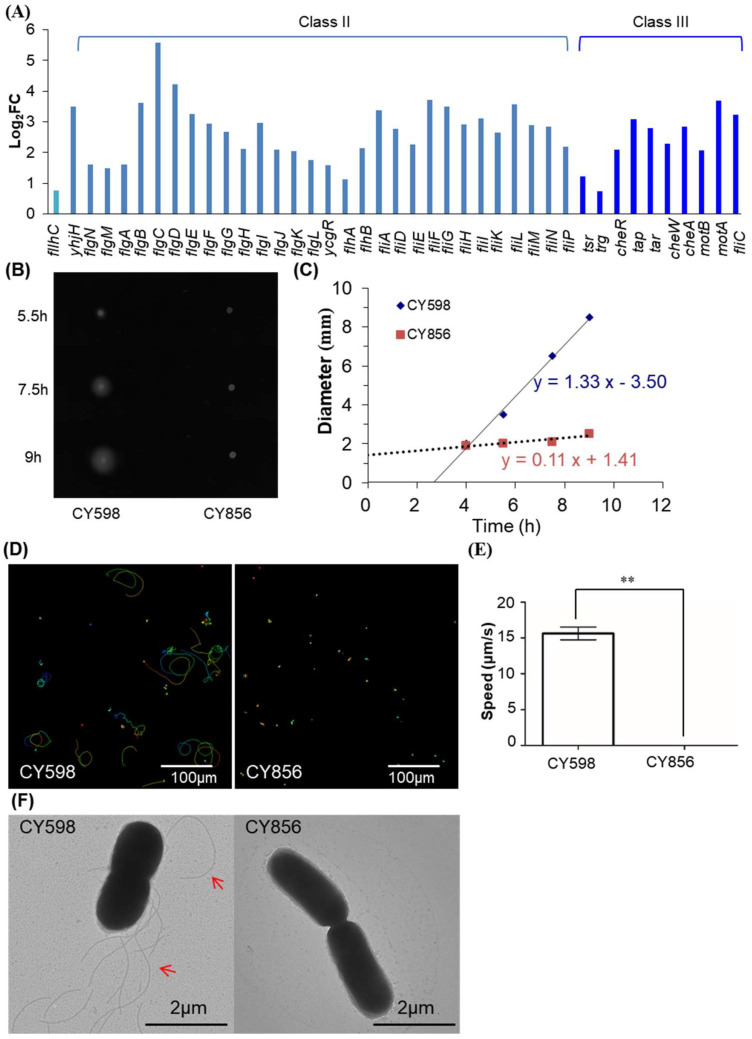
Characterization of Fur regulation on motility. (**A**) Fold changes in DEGs functioning in motility in the transcriptome comparison of NCM3722 versus CY405. The gene of *flhC* belonged to class I in the hierarchy of the flagellar cascade. (**B**) Swimming zones of the two strains on soft-agar plates after incubation at 37 °C for different lengths of time. CY598 is the Fur wild-type strain of NCM3722 with the mutated *fliC* fixed to the wild-type *fliC*. CY856 is the Fur-knockout strain derived from CY598. Two independent experiments were tested, and one representative image is shown. (**C**) Swimming speed of the two strains on soft-agar plates. The diameter of the swimming zone in (**B**) was plotted against time. The slope of the linear fitting indicates swimming speed in unit of mm/h. (**D**) The trajectories of the single-cell movements of the two strains in liquid culture. The rainbow colors correspond to the tracking time. (**E**) Swimming speed of the two strains grown in liquid culture. The swimming speed was expressed as the average ± S.D., which was calculated from 70 to 140 individual trajectories combined from two independent experiments. ** indicates *p* ≤ 0.01 by Student’s *t*-test. (**F**) Flagella of single cells of the two strains. One representative image of two independent experiments is shown. Red arrows mark representative flagella.

## Data Availability

The whole dataset of RNA-seq and ChIP-seq has been deposited to GEO with the accession number of GSE175957.

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
