# Peer review of "Revisiting Fur Regulon Leads to a Comprehensive Understanding of Iron and Fur Regulation"

_ijms, 2023, doi:10.3390/ijms24109078_

Round 1

Reviewer 1 Report

The multidisciplinary study of Hou et al. integrates the results of transcriptomic analysis along with those obtained by ChIP-seq assays to gain new insights regarding the regulatory role of Fur in Escherichia coli iron homeostasis and iron-dependent pathways. Based on their results, the authors reveal 38 new genes directly regulated by Fur. Moreover, they also claim that repression and activation by Fur correlates with higher and lower affinity of this transcriptional factor to the corresponding regulatory regions, respectively.

Although the overall subject of this study and the results obtained are interesting and potentially provide new insights, I am afraid that the data presented in this manuscript are not solid enough and lack sufficient novelty to offer significant advance in the field. In addition, some experimental details are overlooked and missing, which makes it difficult to reproduce and evaluate experimental data by others.

Major points

1. Table S1 contains a list of known (previously reported) and new (reported by Hou et al.) genes.

Unlike the parameters obtained for many previously reported genes, the peak fold enrichment and fold changes (FC) of many transcripts corresponding to newly discovered targets (e.g., dapB, leuD, dinB, pnuC etc.) are rather low, thus raising some doubts regarding the true involvement of Fur in their regulation. The authors should use EMSA to verify that Fur directly binds to the corresponding regions.

2. The authors point to a group of the Fur-dependent genes (p. 13) involved in cell motility and demonstrate that inactivation of Fur impairs the cell’s swimming ability and leads to the loss of flagella (Figure 7). Although the results are interesting, they are not novel!  

The role of Fur in regulation of cell motility has already been studied by Niu et al. (Niu L, Cai W, Cheng X, Li Z, Ruan J, Li F, Qi K, Tu J. Fur Protein Regulates the Motility of Avian Pathogenic Escherichia coli AE17 Through Promoter Regions of the Flagella Key Genes flhD. Front Vet Sci. 2022 Apr 18;9:854916.2) and very similar conclusions and experimental evidence were presented.

 3. The authors speculate that Fur-dependent repression and activation of genes is associated with stronger and weaker interactions of this transcription factor with its corresponding binding sites.  However, these data cannot be substantiated by the experimental data presented in Fig. 3. Instead, the author should consider using EMSA assays to directly compare affinity of Fur to its binding sites within the repressed and activated genes!

4. Many experimental details are missing in Materials & Methods and therefore should be included in the revised manuscript. In addition, the authors should include a section dealing with the statistical analysis.

5. Finally, the text creates a wrong impression that the work presented by Hou et al. is the only study dealing with the Fur regulon in a systematic manner. In fact, similar or alternative approaches have already been employed by other groups to define the core of the E. coli Fur regulon (see ref. 2, 3 and 10). I strongly recommend to make it clear in the revised manuscript.

Reviewer 2 Report

The manuscript by Hou et al. provides an interesting large-scale analysis of the Fur regulon in E. coli. The results expand the previous list of Fur regulated genes and provides evidence that the iron-response in E. coli is apparently exclusively mediated by Fur. The paper is very clear and easy to read. The conclusions are supported by the data. There are just minor points that need to be addressed here.  

1) The introduction needs to be expended. A complete description of the complex regulatory modes by the Fur protein is missing. Fur is able to repress and activate genes, directly and indirectly, both in iron-dependent and -independent manners. Repression occurs by binding to the - 35 and - 10 promoter region, which blocks the access of RNA polymerase while activation is mediated by binding to extended sites between 60 and 240 bp upstream of the transcription start site. Gene expression is exerted either by stimulation of transcription by recruitment of RNA polymerase or by transcriptional silencing by the histone-like nucleoid-associated protein, H-NS. Fur also activates gene expression indirectly via regulation of the small RNA RyhB. Finally, the apo form of Fur also exhibits DNA binding activity and can both repress and activate gene expression. All these information are missing in the introduction. They are needed for the interpretation of the results.

2) p.2 line 3: “biofilm development [2].However, given” a space is missing after the dot

3) p.2 line 83: “Since iron is the cofactor of Fur and Fur is much functional with iron…” and also p.4 result part 2.2. Many genes are detected in both Fe+ and Fe- conditions. This suggests that iron depletion is not be sufficient to fully abrogate the iron-dependent repressor activity of Fur and/or several genes are regulated by apo-Fur. This should be discussed.

4) p.4 bottom, p.6 line 146 and throughout the manuscript: “Using the parameter of fold enrichment as a proxy of the binding strength of Fur” “…the binding strength of Fur on the gens repressed by Fur was significantly stronger…” Gene expression fold-change is an intrinsic property of the gene itself, while binding strength refers to the affinity of Fur for its DNA sequence, which might not be related to the fold changes in gene expression. Please clarify this point.

5) p.4 line 102: “Combining the two ChIP-seq results obtained in the growth conditions with or without iron, 596 binding sites of Fur on the chromosome were identified”. The proportion of genes indirectly regulated by Fur via RhyB should be discussed here. These genes should not come-up by ChIP-seq and therefore the number of genes found by ChIP-seq is expected to be lower than by RNA-seq. Surprisingly, a higher number of genes were found by ChIP-seq (or very close to RNAseq). This should be discussed.

6) p.6 line 150: What are the average positions of the sequences in the genes repressed and activated by Fur ?

7) p.6 line 152: “a most strictly reverse complementary sequence with AT rich”. Unclear what “a most strictly” means. Please revise wordiness.

8) Discussion: The putative role of apo-Fur is not discussed here, neither in the results section (see point 3), this should be considered for the discussion.

Round 2

Reviewer 1 Report

The authors included additional experimental data and text modifications to further support their findings and provide the missing information. Although the overall quality of the manuscript was clearly improved, in my opinion, some corrections are still needed as detailed below.  

-       The authors have substantially revised Fig. 3 by including binding peaks and results of EMSA in panel A and B, respectively. However, the group of transcripts presented in panel A is different from that characterized in panel B. I strongly recommend to improve the consistency and clarity of this figure by making sure that the data presented in both panels of this figure correspond to the same group of transcripts.

-       In some places (e.g., see p. 1, line 44; p. 2, line 91 etc.), E. coli is not in italics.

-       P. 18, line 547: Can the authors explain the choice of sorbitol (for example, instead of more common sugars such as glucose) as a carbon source?

-       P. 20, lines 613 and 615: His- or 6His?

-        P. 20, line 624: the dye used for staining should be specified
